# Characteristics of Buffalo Farming Systems in Turkey Based on a Multivariate Aggregation of Indicators: A Survey Study

**DOI:** 10.3390/ani12213056

**Published:** 2022-11-07

**Authors:** Nursen Ozturk, Omur Kocak, Arzu Peker, Lorenzo Serva, Ferhan Kaygisiz, Pembe Dilara Kecici, Hulya Yalcintan, Halil Ibrahim Kilic, Luisa Magrin

**Affiliations:** 1Department of Animal Breeding and Husbandry, Faculty of Veterinary Medicine, Istanbul University-Cerrahpasa, 34500 Istanbul, Turkey; 2Department of Animal Health Economics and Management, Faculty of Veterinary Medicine, Ankara University, 06110 Ankara, Turkey; 3Department of Animal Medicine, Production and Health, University of Padova, 35020 Padova, Italy

**Keywords:** buffalo farms, multivariate analysis, economic efficiency, intensive management

## Abstract

**Simple Summary:**

In the small–medium scale farm management systems, farmers make rational decisions by combining available inputs to optimize or maximize their production outcome. As the availability of inputs differs in each location, variability in farm typology is becoming more evident, resulting in different economic performances. We aimed to determine buffalo farms’ heterogeneity and differences in productivity and profitability in Marmara Region, Turkey. Six components were identified as the best predictors of farms’ efficiency: “Milk productivity”, “Economic efficiency”, “Roughage management”, “Dual purpose farming”, “Concentrate supply”, and “Fodder production”, each of which aggregate indexes that characterize feeding management and economics, in addition to location, investment in buildings and equipment, and grazing type. These factors’ scores allow for farms’ characterization and identifying management inefficiency areas to prompt early and effective interventions by farmers.

**Abstract:**

This study aimed to determine the heterogeneity that exists in water buffalo husbandry systems in Marmara Region, Turkey. A questionnaire containing a total of 60 indicators was submitted to 52 farmers. A Principal Component Analysis was performed to reduce original variables into a simplified and latent structure, which was characterized by six orthogonal components: milk productivity, economic efficiency, roughage management, dual-purpose farming, concentrate supply, and fodder production. An ANOVA model was applied to the six components to investigate the effects of the province, investment levels, grazing type, milk production, and profitability. Differences in milk productivity, roughage management, concentrate supply, and fodder management were significant according to the province and grazing type, which indicated a difference in intensification levels among the cities. Economic efficiency and dual-purpose farming differed significantly for milk production levels as well as milk productivity, and economic efficiency differed for profitability levels. We found a tendency regarding the impact of roughage management on profitability. The results conclude that profitability was associated with improving the milking traits of buffaloes and roughage management of the farms.

## 1. Introduction

In Europe and Eurasia, Turkey is the second-highest buffalo milk producer, accounting for 21.7% of the total production after Italian farms [1]. The annual milk yield per buffalo in Turkey is 800–900 L on average under extensive farming conditions. This is less than half of the average production of Italian farms [2]. In Italian farms, the high production is obtained with specialized breed under intensive conditions [3,4,5]. In contrast to Italian farms, Turkish buffalo farming is conducted in pasture-based systems by small/medium scale family farms with low-yield Anatolian breeds. Also in Italy, the milk is entirely devoted to pasta filata cheeses production [6,7], but in Turkey the milk is processed into yogurt or kaymak (milk cream) [8]. In 2018, the ratio of the numbers of buffalo to the total number of large ruminants was reported to be 1.0%. However, the reported buffalo milk production was only 0.3% of that of large ruminants [9]. A national project involving eighteen cities began in 2011, which aimed to improve the milk traits of the Anatolian Buffaloes through selection by breeders [10]. In the scope of this project, milk yield and days in milk of buffaloes aimed to be improved at 1200 kg/buffalo and 250 days, respectively. As a result of this project, various buffalo farms showed a tendency towards intensification. Farmers started to demand large-scale farms and high-yielding breeding buffaloes, and, consequently, changed their management strategies. For example, they started to provide concentrate feed beside pasture for grazing, use automatic milking machines, include/shift paid labour, as well as increase investment in the extension of farm facilities [11]. Moreover, these farms generated their income by selling raw milk, in contrast to the traditionally managed farms [12].

Although the increasing intensification level of a farm often implies higher milk yield, economic performance, and enhanced profitability [13,14,15], it would not be adequate for all cases [16,17,18]. Poczta et al. [19] revealed that intensive dairy farms in Sweden, Luxembourg, the UK, the Netherlands, Ireland, and Denmark showed high production efficiency and income value but poor profitability indicators. This was most probably due to high production costs and a high value of total assets in relation to incomes. On the contrary, extensive small-scale farms in Portugal, Spain, and Bulgaria had high levels of profitability due to low levels of external input costs—mainly, the hired labour cost.

The farm intensification process and implementation of any new management systems should rely on the analysis of similarities and differences between the current and new farm profiles [20]. In this regard, aggregating multiple indicators can lead to more comprehensive results, which would help to develop more synthetic parameters [21], and finally, distinguishing groups, identifying the weak and strong points, proposing a structural development plan for the sector [22].

In the case of buffalo husbandry, farmers showed high variability in farm management strategies [23] and efficiency levels [12,24]. Therefore, such a typology analysis would be explanatory to determine the heterogeneity that exists in buffalo husbandry. Previous studies point out that pasture use (pasture type and duration); feed production (producing or purchasing feed); diet composition (provided concentrate and forage amount, silage inclusion); labour supply (including family or paid labour); and investments in facilities (including equipment and/or facilities related to milking, feeding and/or labouring) affect the economic performance of a farm, because the rational combination of those indicators makes a farm self-sufficient and efficient [16,21,22,25,26,27,28,29]. Furthermore, the province of a farm factors in the individual farm management differences. These may be due to the similar constraints, opportunities, or cultural background that may influence the farmer’s attitudes [30,31].

This study aimed to identify the sources of variation in the buffalo production system. Further, it aimed to aggregate indicators to establish an efficient decision-making process at farm level. While doing this, land use, herd structure, animal nutrition and production, health status, economic performance, investment in farm building and equipment, and farmer profile (demographic information, i.e., gender and education status) were taken in consideration. The results of this study can help to reconsider and restructure the current support policy of the decision makers by determining the strengths and weaknesses of buffalo farming systems from an economic perspective.

## 2. Materials and Methods

### 2.1. Area of the Study and Farm Selection

Marmara Region is in the north-west of Turkey, and it is formed by three sub-regions (Istanbul (TR-1), the west Marmara (TR-2), and the east Marmara (TR-4)). When choosing the cities representing each sub-region, we considered those with the highest number of buffalos and where the national buffalo breeding project was conducted. As a result, the cities of Istanbul (TR-1), Kocaeli (TR-2), and Balikesir (TR-4) were identified as qualified by the above-mentioned criteria. When choosing the buffalo farms, it was conditioned that 2/3 of the annual income of a farm should be provided by buffalo husbandry. In the study area, the number of buffalo farms in Istanbul, Kocaeli, and Balikesir provinces were 240, 104, and 191, respectively. In total, 60 farmers were surveyed. Two farms were detected as outliers and excluded from further analysis because of performing a highly intensive production system, and thus, increasing the variability of the results. Furthermore, due to some missing answers, 6 farms were excluded. The answers of 52 farms were used as the material of this study, consisting of 25 farms in Istanbul, 17 farms in Kocaeli, and 10 farms in Balikesir.

### 2.2. Data Collection and Processing

Data were collected through face-to-face interviews by the same trained researcher in June and December 2018, which corresponded to the 2017–2018 production period. The farmers answered a set of questions regarding land use, herd structure, animal nutrition, health trait, animal production, economic performance, labour information, investment in farm building and equipment, and farmer profile topics (Study survey can be found in the Appendix A). When calculating the investment in farm building and equipment scores, one degree was assigned for each of the following facilities or equipment if it was owned by the farmer i.e., milking unit, milk-cooling tank, generator, feed-crushing unit, feed-mixing unit, buffalo barn, courtyard, concentrate feed warehouse, roughage warehouse, worker office/lodging. Thus, the investments of farms in the buildings and equipment score were evaluated with a score between 0 and 10. Indicators were presented according to the above-mentioned macro groups, i.e., land use, herd information, animal nutrition information, herd health information, animal production information, economic performance information, labour information, investment information, and farmer profile, and the original values of the indicators used in the further analysis are shown in Table 1 and Table 2.

The indicators that were used in the further statistical analysis were obtained through calculation of the data provided by the questionnaire. The lactation information of the buffaloes was obtained from the National Buffalo Breeding Project. In the scope of the National Buffalo Breeding Project, the milk yields of the lactating buffaloes were recorded in each month; in order to estimate milk production, they were corrected to 305 days.

Concentrates and roughage amount were calculated on an as-fed basis. The feed intake from the pasture consumption was not included in any of the analyses. The weight of an adult female buffalo was reported as between 411 and 518 kg [32]. Calculation methods for the used indicators are provided in Appendix A.

### 2.3. Statistical Analysis

#### 2.3.1. Descriptive Statistics

Descriptive statistics for the original variables reported in Table 1 concern the mean, standard deviation, minimum and maximum values for continuous qualitative variables. Categorical variables (i.e., gender, education status, grazing information, membership of union breeders) are reported as absolute frequencies in Table 2.

#### 2.3.2. Multivariate Analysis

The latent variable extraction using Principal Component Regression (PCR) solves the unwanted collinearity among variables when resolving problems of regression and discriminant analysis [33]. The Principal Components Analysis (PCA) extracts important information from several dependent inter-correlated variables and expresses this information as a set of new orthogonal variables called principal components (PC). In addition, the PCA can be extended to handle qualitative and quantitative variables as Multiple Factor Analysis (MFA) [34]. The MFA is used when several sets of variables have been measured on the same set of observations variables. The MFA works in two steps; the first is intended to normalize each of the individual datasets. In this way the first principal component has the same length. The second step combines these data into a common table (grand table) from a (non-normalized) PCA of the concatenated normalized data [35]. The cluster analysis (CA) is an unsupervised method used to find sub-groups or “clusters” in a dataset, according to their similarities among variables (i.e., PC) [36]. It was used in many herd applications in the characterization and classification of farms [37], usually in terms of their structural [27], techno-economic [13,16,19,38,39], and socio-economic aspects [31].

With the aim of avoiding the presence of highly correlated indicators, before any analysis, all linear dependent variables were evaluated, and many were removed. Secondly, a matrix of correlation was calculated, and indicators showing many high correlations (r > 0.6) were removed from the dataset. The multivariate approach was used to reduce original variables into fewer orthogonal latent variables, and it was calculated in two steps. The first step allowed an exploratory representation of the original variables, which were reduced in the new space of less dimensionality of the multivariate data. This step was completed through the MFA approach, which is used as a visual descriptor of the relationships between original and latent variables and categorical groups. For this step, only the quantitative variables were used for PCA calculations, while the categorical ones were supplementary. The supplementary variables, which are not used to determine the principal factors, allow their coordinates to be visualized using the information provided by the performed PCA on active variables. Finally, the Hierarchical Clustering on Principal Components (HCPC) performs clustering on individuals, combining principal components methods, hierarchical clustering, and partitioning to improve visualization and highlight the similarities between individuals [40].

Furthermore, the second step performed a PCA analysis in the original dataset after it had undergone variable selection, ensuring that the overall Kaiser–Meyer–Olkin (KMO) test for Sample Adequacy was greater than 0.6. The PCA for each variable allows separation of the communality (i.e., the proportion of variance explained by a common factor) from the uniqueness (i.e., the amount of variability that each variable does not share with other variables, and it is 1 -communality). The uniqueness is greater when the relevance of the variable in the factor model is lower [41]. The uniqueness consists of a random error summed to the variable’s specific variance without the possibility of being separated [21]. The “varimax” rotation was applied to the standardized (centred and scaled) variables of the dataset to facilitate the interpretation of the results [42]. The extracted components were determined by the Kaiser criterion (eigenvalues > 1) [43]. Each PC was considered related to a variable if the absolute values of the loading were greater than 0.55 [21].

#### 2.3.3. Analysis of Variance

The new PCs extracted from the rotated PCA were used as variables in each of the 52 farms and evaluated for the normality assumption by the Shapiro–Wilk test, visual inspection of the frequency distribution, and the Q-Q plot (quantile-quantile plot).

The main effects on the newly extracted components were analysed using an ANOVA model applied to the province, investment level, grazing type, milk productivity, and profitability as fixed effects. Post hoc pairwise comparisons were run between factor levels using Bonferroni correction.

All statistical analyses were performed with the use of R version 4.1.3 (10 March 2022), the R Commander R package version 2.7-2 [44], the plug-in FactoMineR 2.4 [45], and package ‘psych’ [46], ‘stats’ [47], and ‘Factor Assumptions’.

## 3. Results

### 3.1. General Characteristics of the Farms

The study area had a temperate climate, in which the average monthly temperature reached a minimum of about 4.6 °C in January 2018 and a maximum of about 24.8 °C in August 2018 [48]. The altitude of the farms ranged between 8 and 308 metres. Some herds used communal grazing, while others used their own land for grazing purposes. Pastureland included meadow, and forest for communal grazing. Areas used for common grazing are not private; rather, they are open for public use. Farmers in Kocaeli used their own land for grazing purposes. However, the farmers stated no special pasture treatments to improve pasture quality. During the summer, farmers provided concentrate, grains, and roughage besides pasture to the lactating buffaloes. A typical diet for a lactating buffalo consisted of concentrate feed supplemented by grains (mainly wheat, wheat shorts, wheat bran, and beet pulp). Wheat straw, barley straw, alfalfa hay, and corn silage were supplied as roughage sources. In terms of diet composition, some regional differences occurred. Farmers in Istanbul used red dog (fine bran) as a source of wheat, while cottonseed meal was only used by farmers in Balikesir. All farms used indigenous Anatolian Buffalo breed and milked twice a day. The majority of the farms used automatic milking machines (57.7%). Artificial insemination was not a common practice, but the buffalo bulls were used for insemination. Instead of owning buffalo bull(s), some farms in Balikesir shared bull(s) during insemination season. Diarrhoea was the most frequently reported disease in buffalo calves during winter, followed by mastitis. Abortion was reported by 23 farms (Appendix A).

Most of the farm owners were male. The age of the farmers was on average 48 years old, and the experience was 28.7 years. All the farmers were members of a breeding union. The average number of female breeding buffaloes, buffalo bulls, and buffalo calves was determined as 30.0, 1.6, and 16.4 heads, respectively. Due to the grazing period, a decrease in the amount of feed supply was observed between the fall–winter and spring–summer seasons. Concentrate supply decreased from 6.1 to 5.3 kg, while roughage supply decreased from 10.1 to 5.6 kg. The proportion of silage in the diet decreased from 28.7 to 18.4%. Moreover, the average pasture use was determined as 92.6 h in fall–winter and 270.9 h in spring–summer. On average, the daily milk yield of a lactating buffalo was 4.9 L and days in milk was 244.3 days. Milk selling was the main income generating activity (USD 24,696.4), followed by yogurt (USD 4310.1) and kaymak (USD 2275.4). The average amount of total labour was 2.5 annual work units (AWU), and the majority of the workforce was composed of family members (Table 1).

Farmers showed different characteristics for some of the management practices. It was determined that some of the farmers did not apply the following practices: animal feed production, owning a buffalo bull, pasture use in fall–winter, providing concentrate and/or roughage feed in spring–summer, silage in diet, feed purchase, culling rate, income activity and product prices, and using paid and family labour. Therefore, the minimum values for those indicators resulted in 0 values. The percentage of farms with 0 values corresponds to each indicator can be found in Appendix A. 

### 3.2. Results of Multivariate Analysis of Calculated Indicators

The MFA reduces the number of initial variables into fewer and better interpretable features (dimension), and the first five dimensions had eigenvalues > 1, explaining 59% of the variance. The eigenvalues of the MFA ranged from 1.2 to 2.7 (Table 3). The HCPC cluster split farms into three classes, which were relatively similar to their region. Most of the farms in Kocaeli were clustered in cluster 1, farms in Balikesir were clustered in cluster 2, and those in Istanbul were clustered in cluster 3 (Figure 1).

The PCA was performed on the continuous variables’ dataset after removing those with KMO < 0.6, in order to ensure that the final KMO for the model was > 0.6. The kept variables were 33 and were marked in Table 1. After the varimax rotation, the first 10 rotated components (RC) had eigenvalues > 1 and explained 82% of the cumulative variance (14, 12, 12, 11, 9, 8, 8, 4, 4, 4% of explained variance for RC −1, −2, −4, −5, −6, −7, −3, −8, −10, and −9, respectively). The communality for the original variables was reported in Table 4.

The varimax rotation of extracted components provided six groupings among factors. In cluster 1, the number of buffalo calves (67% loading), days in milk (75% loading), average milk production in fall–winter (78% loading), average milk production in spring–summer (76% loading), and paid labour/total labour (90% loading) were positively loaded, and family labour/total labour (−90%) was negatively loaded. This cluster was defined as “milk productivity”.

Cluster 2 was positively associated with the highest number of lactating buffalo/total number of breeding buffalo (72% loading), gross profit/AWU (56% loading), gross profit/breeding buffalo (89% loading), gross profit/1 L of milk production (74% loading), and negatively associated with concentre feed offer per 1 L of milk production (−63% loading), as well as variable costs/1 L of milk production (−60% loading). Therefore, it was labelled as “economic efficiency”.

Cluster 3 was negatively associated with roughage cost/breeding buffalo (−83% loading) and roughage cost/1 L of milk production (−67% loading), and it was positively associated with milk price (60% loading). This cluster was described as “roughage management”.

In Cluster 4, gross production value/1 L of milk production (84% loading) and fattening calves’ income/dairy income (95% loading) were positively associated, and the share of dairy income in total income was negatively associated (−93%). This cluster was described as “dual-purpose production”. Gross profit/1 L of milk production was double-loaded in cluster 2 and cluster 4 with factor loadings of 74% and 54%, respectively. Due to the higher loading, we considered this indicator in cluster 2.

Cluster 5 was positively associated with concentrate feed offer/roughage offer in fall–winter/lactating buffalo (53% loading), variable costs/breeding buffalo (84% loading), concentrate cost/breeding buffalo (94% loading), and concentrate cost/1 L of milk production (65% loading). This cluster was defined as “concentrate supply”.

Cluster 6 was positively associated with land used for roughage production (88% loading) and land used for roughage and grain production per breeding buffalo (83% loading). The purchased-to-total feed cost (−50%) indicator contributed negatively to cluster 6. This cluster was defined as “fodder production” (Table 4).

### 3.3. Results of Variance Analysis of Clusters Related to Farm Characteristics

According to the results of the ANOVA, the milk productivity (*p* < 0.001), roughage management (*p* < 0.001), concentrate supply (*p* < 0.01), and fodder production (*p* < 0.001) differed significantly according to the province. Compared to their counterparts, farms in Istanbul had higher scores on milk productivity. Farms in Kocaeli had higher roughage management scores. Moreover, farms in Istanbul and Balikesir had higher scores on concentrate intake, and farms in Kocaeli had higher scores on fodder production than farms in Istanbul.

An increase in the investment level of the farms resulted in higher scores for milk productivity. Farms with high investment levels had higher scores on milk productivity than low- and medium-scored farms (*p* < 0.001).

Milk productivity (*p* < 0.01), roughage management (*p* < 0.001), concentrate supply (*p* < 0.01), and fodder production (*p* < 0.001) differed significantly according to the grazing type. Farmers using common grazing land for the grazing purposes had high scores on milk productivity and concentrate supply. On the contrary, they had lower values for roughage management and fodder production.

Milk productivity (*p* < 0.001), economic efficiency (*p* < 0.05), and dual-purpose production (*p* < 0.001) differed significantly according to the milk production levels. In parallel to the increase in milk productivity, farms had higher scores on milk production. Farms with high milk production had higher economic efficiency scores than farms with low milk production, and those with low scores on dual-purpose production than low- and medium-level producers.

The effect of profitability on milk production (*p* < 0.001) and economic efficiency (*p* < 0.001) was found to be significant. Farms with high and medium profitability levels had higher production levels, and those with an increase in profitability and economic efficiency scores were observed to increase (Table 5).

## 4. Discussion

Typology studies are helpful tools to use for defining farm characteristics. According to the identified weaknesses of the farms, it is possible to prioritize plans and specific research or development policies leading to the generation and adaptation of technologies suitable for each system [16]. In this study, a PCA was used to determine the heterogeneity among the buffalo farms operating in the Marmara Region. Our results revealed that most of the differences among farms were caused by the “milk productivity”, “economic efficiency”, “roughage management”, “dual purpose production”, “concentrate supply” and “fodder production” levels.

The results of the HCPC analysis showed that most of the buffalo farms were clustered similar to their location, which verified one of the study hypotheses that province would be a determinant for the farm management differences. In the study, some management practices, i.e., pasture use, diet composition, marketing, and labour force were observed to be divergent for the three provinces. This result was due to biophysical and socio-economic conditions. Gökdai et al. [31] also revealed that Turkish and Italian goat farms tend to cluster correspondingly to their location.

Turkish buffalo husbandry was reported to be in the process of intensification [11]. Consequently, our results revealed that the milk productivity component—which is one of the reference points for the intensification—explained the highest variation by 14% (Table 4). Aside from the traditional management systems, we identified some farms employing a high number of younger generations, experiencing improved milk production levels and lactation duration, and preferring to include paid labour in the production chain instead of the family work force. The intensification process of a small-scale farm begins with a farmer “response”, which is subject to various drivers. Population dynamics, the dietary shifts of the consumers, market access, input/output prices, land opportunity cost, labour cost, the socio-cultural traits of farmers, and technology accessibility were reported as determinants that may affect the farmer’s decision about intensification [49]. When there is an increase in consumer demand for buffalo milk, farmers tend to improve milk yield, extend lactation duration, and thus, supply buffalo milk to meet consumer demand [50]. Compared to other provinces, the high consumer demand for buffalo milk in Istanbul would motivate farmers to improve their milk traits, leading them to increase the intensification level. Moreover, due to the easy market access, farmers in Istanbul had opportunities to either sell raw milk directly to consumers or to milk processing chains. However, farmers in Kocaeli and Balikesir stated that there was no consumer demand for raw milk, which required them to process their milk into yogurt or kaymak. Lastly, another determinant of the milk productivity component was found in the number of paid labourers. The main tasks applied in a daily routine, i.e., calf management, preparing feed and feeding animals, milking, bedding, and the cleaning of stalls and indoor alleys become evidently significant with an increase in farm size [51]. Therefore, in parallel to the increase in the milk productivity in Istanbul, we observed that buffalo farms shifted their labour preference to paid workers, and, most probably, production transforms into a labour-intensive system.

Feeding strategies provide solid information about the farm management system [27]. In the study, we determined three components that were related to feeding strategies in buffalo husbandry in Marmara Region. “Roughage management”, “concentrate supply”, and “fodder production” were identified as variation sources, which together explained 29% of the variation. The results of the ANOVA showed a significant effect of the province on “roughage management”, “concentrate supply”, and “fodder production” components. Farms in Istanbul were identified as having better scores on “roughage management” than the other provinces. This is because that component was negatively associated with roughage cost per female buffalo and per 1 L of milk production. A negative association with the cost indicators indicated better input use. In other words, lower scores from this component indicated better performance. “Concentrate supply” was positively associated with concentrate-to-roughage ratio in fall–winter, concentrate cost per female buffalo and per 1 L of milk production, and variable costs per female buffalo. Due to the limited pasture use and metabolizable energy provided by the pasture, in fall–winter, milk production relies heavily on the concentrate and ensiled forage [52,53]. Therefore, farmers should provide more concentrates if they produce milk outside of the spring–summer season. However, for this component, farms did not only differ by their concentrate supply in fall–winter. The positive loadings of the concentrate cost (per female buffalo and 1 L of milk production) and variable cost (per female buffalo) indicators would indicate that farms with high value in this component may supply a high amount of concentrate throughout the year. In this study, the results of the ANOVA showed that farms in Istanbul had higher scores on concentrate supply, which is probably linked to their high milk productivity level. Due to the implementation of more intensive production systems contributing to an increase in dairy milk production [54], the nutritional demands of the animals are met by supplying a high amount of concentrate [55]. Compared to the other cities, it was determined that farms in Istanbul apply an intensive management strategy.

Another important heterogeneity among farms was identified in “fodder production”. This component was significantly explained by the land use for roughage production, land used for roughage and grain production per female buffalo, and the share of purchased feed cost in total feed cost indicators. Farms in Istanbul had lower scores than farms in Kocaeli for this component. In parallel to the milk productivity level, Istanbul farmers may use increased external feed resources in their diet composition, affecting the share of purchased feed in total feed cost. Moreover, land use opportunity may exist in this context. In Istanbul, land is more valuable due to urbanization. Farmers may prefer to use their land for another activity rather than animal nutrition, which generates more income [56].

Investment in farm buildings and equipment provides modernization and sustainability of production. By implementing milking technology, hygienic milking can be sustained, and technologies in animal nutrition enhance the efficient use of feed resources [16]. In this study, it was possible to record when farmers implemented the following investment criteria, i.e., milking unit, milk-cooling tank, generator, feed-crushing unit, feed-mixing unit, buffalo barn, courtyard, concentrate feed warehouse, roughage warehouse, and worker office/lodging. The results of the ANOVA revealed that the farms that invested most in their production had the highest milk production values. Certainly, high milk producers require advanced milking and feeding systems, and, as well as due to their labour-intensive nature, they need a worker office/lodging. Gelasakis et al. [27], Martinez et al. [38], and Lapple and Sirr [57] revealed that the investment in animal production is one of the determinants of intensification level. Therefore, we would like to point out that buffalo farms with high milk productivity as well as a high investment level would be in the process of being intensified.

In this study, we identified that buffalo husbandry in Marmara Region was based on grazing. Two types of grazing were performed by the farmers, namely common grazing (meadow and/or forest) and own-land grazing. Most of the farms in Kocaeli used their own land for grazing purposes because of the lack of natural pastures, whereas farmers in Istanbul and Balikesir used common pastures. Own-land grazing was performed by sending animals to their land after the harvest (grain and/or roughage) or to non-cultivated land. We should mention that none of the farmers stated a specific pasture treatment to improve pasture quality, neither for common grazing nor for own-land grazing systems. Therefore, the significant results of the effect of grazing type on milk productivity level, roughage management, concentrate supply, and fodder production most likely occurred due to a city-specific grazing type. Several authors have suggested that when a pasture is efficiently managed, the feed cost of a farm decreases and profit increases significantly [58,59]. However, we did not identify any significant effect of the grazing type on the economic efficiency component.

Milk productivity, economic efficiency, and dual-purpose farming differed significantly for the milk production levels. The results showed that the increase in milk production improved economic efficiency. In our study, the “Economic efficiency” component was positively associated with the highest number of lactating buffalo/total number of female breeding buffalo, gross profit/female buffalo, and gross profit/1 L of milk production indicators, while it was negatively associated with concentrate offer/1 L of milk production and variable cost/1 L of milk production indicators. Economic efficiency is defined as a farm’s ability to produce maximum output with a certain amount of input while allocating inputs with a certain cost in an optimal way. To achieve economic efficiency, farms should (i) improve output levels and (ii) use inputs at an optimal cost and/or amount [60]. Consistent with the above-mentioned definition, it was determined that a high amount of milk producers in relation to their production cost was more economically efficient than a low amount of milk producers. “Dual-purpose farming” was positively associated with the fattening calves’ income/dairy income and gross domestic production value/1 L of milk production, and negatively associated with the share of dairy income in total income indicators. In the study area, some farms diversified their income by performing other activities.

From the previous literature, it appears that the profitability of a dairy farm is mainly linked to various factors that are unique in their management systems and regions [13,15,16,17,19]. When a farm is dependent on external inputs, operates less efficiently in relation to cash input-use relative to the value of production, and has poor internal efficiency, profitability decreases and vice versa [15,57]. In our case, buffalo farms producing a high amount of milk were more profitable when they operated in pasture-based systems. We did not find a significant relationship between profitability and roughage management, but there was a tendency for profitable farms to have better roughage management. In the study, farms with lower roughage cost per buffalo and per L of milk production tended to be more profitable. Feed cost has the highest share in the production cost of a farm, and when the farm has the ability to efficiently manage its feeding management, its profitability can increase. Moreover, the milk production potential of buffalo is not very high, and farmers used more roughage than concentrate (a diet consisted of 60% roughage on average) for animal nutrition. At this point, farms who can successfully manage forage may see their profitability improve. We point out that the feeding strategy is a significant variation source among pasture-based buffalo farms. Therefore, it is important to identify the different pasture management and fodder-conservation techniques, as well as their relationship with the farm’s economic performance.

Lastly, our analysis enabled us to compare performances at farm level (Figure 2a) and territorial level (Figure 2b) in relation to the extracted factors in the spider charts. When comparing the performances of randomly chosen farms, farm 27 showed an interesting pattern in obtaining high economic efficiency scores. In this case, having a good roughage management, diversifying farm activities, and producing own feed could lead to improved economic efficiency scores. Similarly, in the case of territorial level, farms in Istanbul showed intensification characteristics compared to other regions, in which they obtained higher scores on milk productivity and concentrate intake, and low scores on fodder production. However, they need to improve their roughage management, farm activities, and feed production; this could improve their economic efficiency, as in the example of farms located in Kocaeli. Those farms had slightly better economic efficiency scores than farms in Istanbul, and they had the highest roughage management scores as well as fodder production scores. Atzori et al. [21] reported the benefits of using spider charts, as they quantify the farms needing improvement in a specific area. Moreover, they are a solid tool for supporting short-term decision making when a complete balance sheet is not available. Spider charts provide recommendations for improving performance within specific areas by facilitating the comparison with other farms.

## 5. Conclusions

Milk productivity, feed management, and economic efficiency were identified as the main variation sources among buffalo farms operating in Marmara Region. The regional differences occurred in terms of farm management practices. Farms in Istanbul had higher milk productivity, offered a higher amount of concentrate, and were more dependent on external feed resources, which depicts that they were in the process of intensification. However, our spider chart showed that those farms needed to improve their economic efficiency compared to Kocaeli farms, which successfully managed roughage supply and were independent regarding the fodder production.

We would like to underline that from an economic point of view, concentrate supply may not be as important as roughage management due to the low milk production capacity of the buffaloes, because milk productivity would not correspond to concentrate intake. This issue would be one of the reasons for low economic efficiency and profitability scores in some cases. The ANOVA analysis showed that there was a tendency for an increased profitability of farms which efficiently manage their roughage supply. Therefore, supporting buffalo farms to intensify their production is essential, but farmers should consider sustainable intensification in which pasture is being efficiently used, and input use is less dependent on external sources. Furthermore, since buffalo grazing is performed in pastures/lands without specific pasture treatment, the improvement in the efficiency of pasture use is essential.

Since regional differences occurred in terms of intensification and feed management practices that caused different economic efficiencies among the farms, local support policies are essential to improve specific areas. This approach allows the farmer association and public government to easily analyse the buffalo livestock sector and formulate a better support policy. Creating an online accessible platform will help farmers to input their own data and understand their strengths and weaknesses. Moreover, the approach could be replicated in other dairy husbandry contexts.

Although typology studies provide a “snapshot of farm situation” in a certain period, timely studies provide useful information to understand the inefficient areas that need to be identified and urgently supported.

## Figures and Tables

**Figure 1 animals-12-03056-f001:**
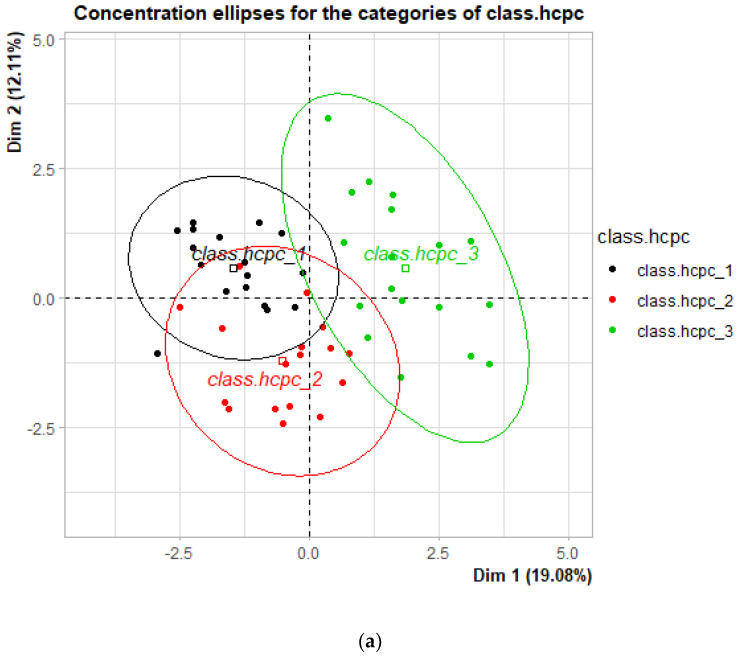
(**a**) The location of each cluster was revealed according to the hierarchical cluster analysis (class.hcpc). Dots represent the coordinates of each farm, squares represent the geometric centre of each cluster, circles indicate the 95% confidence interval, dim 1 and 2 are the first two dimensions obtained from the multiple factor analysis (MFA), and in brackets are reported the explained variances from each dim. (**b**) The location of each cluster was revealed from the region (Q1) where Koc. refers to Kocaeli, Bal. refers to Balikesir, Ist. refers to Istanbul, dots represent the coordinates of each farm, squares represent the geometric centre of each cluster, circles indicate the 95% confidence interval, dim 1 and 2 are the first two dimensions obtained from the multiple factor analysis (MFA), and in brackets are reported the explained variances from each dim.

**Figure 2 animals-12-03056-f002:**
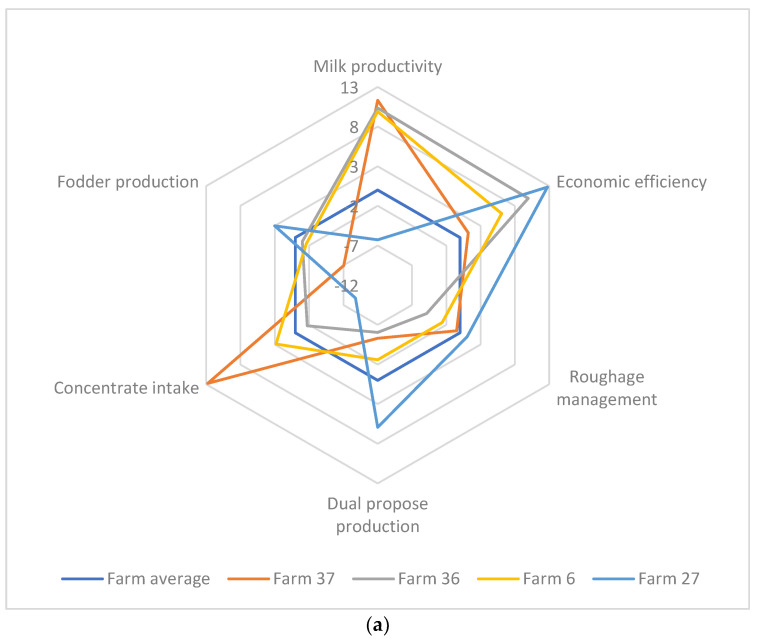
(**a**) Comparison of the randomly selected farm performances in relation to the extracted factors obtained from the PCA. Performance of a farm should be compared with the average farm. (**b**) Comparison of the territorial performances in relation to the extracted factors obtained from the PCA. In the chart, BAL. represents Balikesir, IST. Represents Istanbul, and KOC. represents Kocaeli.

**Table 1 animals-12-03056-t001:** Structural and management characteristics of the buffalo farms in Marmara region (mean ± standard deviation, minimum, maximum).

Indicators	Mean ± Standard Deviation	Minimum	Maximum
Land-use information			
Land used for roughage production, ha *^,1^	64.8 ± 66.2	0.0	275.0
Land used for grain production, ha *^,2^	37.4 ± 61.9	0.0	300.0
Land used for roughage and grain production per breeding buffalo, ha *^,1^	4.3 ± 3.7	0.0	17.9
Herd information			
Number of female breeding buffalo, n	30.0	8.0	102.0
Highest number of lactating buffalo/Total number of female breeding buffalo *^,1^	0.71 ± 0.18	0.22	1.0
Number of buffalo bull, n	1.62 ± 1.09	0.0	5.0
Breeding buffalo/Buffalo bull *^,1^	19.3 ± 12.3	0.0	55.0
Number of buffalo calves, n *^,1^	16.4 ± 10.4	3.0	51.0
Age at first mating, month	24.6 ± 6.0	14.0	48.0
Animal nutrition information			
Amount of daily concentrate feed offer per lactating buffalo in fall–winter, kg ^§^	6.1 ± 3.2	1.3	19.5
Amount of daily concentrate feed offer per lactating buffalo in spring–summer, kg ^§§^	5.3 ± 3.2	0.0	19.5
Amount of daily roughage offer per lactating buffalo in fall–winter, kg	10.1 ± 4.8	1.6	21.9
Amount of daily roughage offer per lactating buffalo in spring–summer, kg	5.6 ± 4.7	0.0	19.9
Concentrate-to-roughage ratio in fall–winter *^,1^	0.77 ± 0.55	0.09	2.9
Concentrate-to-roughage ratio in spring–summer *^,1^	1.0 ± 1.1	0.0	5.0
Total amount of concentrate feed offer per breeding buffalo, kg	2060.5 ± 1101.0	270.0	7020.0
Concentre feed offer per 1 L of milk production, kg *^,1^	2.5 ± 1.2	0.25	6.3
Total amount of roughage offer per breeding buffalo, kg	2871.8 ± 1422.8	865.7	7170.0
Roughage offer per 1 L of milk production, kg *^,1^	3.8 ± 2.6	1.2	15.7
Total annual pasture use, hour *^,2^	2181.0 ± 713.4	859.0	4280.0
Average monthly pasture use in fall–winter, hour	92.6 ± 61.5	0.0	243.4
Average monthly pasture use in spring–summer, hour	270.9± 69.4	143.2	510.0
Silage in diet, winter, % *^,2^	28.7 ± 23.1	0.0	78.7
Silage in diet, summer, % *^,2^	18.4 ± 25.8	0.0	100.0
Herd health information			
Culling rate *^,2^	0.06 ± 0.07	0.00	0.30
Animal production information			
Daily milk yield, buffalo/lactation, L *^,1^	4.9 ± 0.9	2.6	6.8
Amount of total milk production, herd/year, L	26,687.8 ± 21,078.4	6410.2	89,210.0
Days in milk, d *^,2^	244.3 ± 28.9	180.0	299.3
Average daily herd milk production in fall–winter, L *^,1^	67.2 ± 54.8	6.9	240.8
Average daily herd milk production in spring–summer, L *^,1^	2405.0 ± 1981.0	418.1	8508.3
Weaning age of buffalo calves, day	115.4 ± 49.1	60.0	270.0
Economic performance information ^†^			
Milk income, USD *^,1^	24,696.4 ± 29,068.1	0.0	138,557.1
Yogurt income, USD *^,2^	4310.1 ± 9994.6	0.0	55,881.5
Kaymak income, USD *^,2^	2275.4 ± 6995.0	0.0	37,827.8
Milk price per kg, USD *^,2^	0.92 ± 0.47	0.0	1.6
Yogurt price per kg, USD *^,2^	0.31 ± 0.59	0.0	1.9
Kaymak price per kg, USD *^,2^	1.8 ± 4.2	0.0	14.0
Share of dairy income in total income, % *^,1^	0.63 ± 0.16	0.32	0.93
Gross production value per year, USD	48,355.2 ± 33,388.2	9440.4	157,907.8
Gross production value/Breeding buffalo, USD *^,2^	1752.1 ± 880.1	193.0	5778.5
Gross production value/1 L of milk production, USD *^,1^	0.40 ± 0.12	0.23	0.75
Variable costs per year, USD	30,538.9 ± 24,159.0	4176.6	119,127.8
Variable costs/Breeding buffalo, USD *^,2^	1002.0 ± 365.6	263.9	2019.1
Variable costs/1 L of milk production, US$ *^,2^	0.25 ± 0.10	0.08	0.60
Concentrate costs/Breeding buffalo, USD *^,2^	478.5 ± 234.5	22.1	1226.9
Concentrate costs/1 L of milk production, USD/L *^,1^	0.12 ± 0.06	0.01	0.30
Roughage costs/Breeding buffalo, USD *^,2^	323.7 ± 190.1	38.6	778.0
Roughage costs/1 L of milk production, USD *^,1^	0.08 ± 0.05	0.01	0.21
Gross profit/AWU ^#^, USD *^,1^	1776.3 ± 2161.7	216.3	14,604.3
Gross profit/Breeding buffalo, USD *^,2^	626.5 ± 490.3	96.9	2358.9
Gross profit/1 L of milk production, USD *^,2^	0.15 ± 0.09	0.03	0.46
Purchased feed cost/Total feed cost *^,1^	63.2 ± 25.5	0.0	100.0
Fattening calves’ income/Dairy income, % *^,1^	43.8 ± 42.9	0.0	151.6
Labour information			
Total labour, AWU	2.5 ± 1.6	0.33	10.0
Family labour/Total labour, AWU *^,1^	0.72 ± 0.38	0.0	1.0
Paid labour/Total labour, AWU *^,1^	0.28 ± 0.38	0.0	1.00
Total labour/Breeding buffalo, AWU *^,2^	0.11 ± 0.08	0.01	0.47
Investment information			
Investments in buildings and equipment score (0–10)	6.3 ± 2.6	0.0	10.0
Farmer profile			
Age, year	48.0 ± 11.8	32.0	74.0
Experience, year	28.7 ± 13.6	3.0	60.0

^†^ Data were collected in Turkish currency (TL); before the analysis, all the financial parameters were converted to US dollars. During the study period (2017–2018), USD 1 = TL 4.99. ^§^ Fall–winter includes January, February, March, September, October, November, and December. ^§§^ Spring–summer includes March, April, May, June, July, and August. * Variables included in the multivariate factor analysis. ^1^ Variables kept for the principal component analysis. ^2^ Variables excluded from the principal component analysis. Variables without an indication were previously excluded from the analysis due to their high linear correlation. ^#^ AWU: annual working unit.

**Table 2 animals-12-03056-t002:** Qualitative structural and management characteristics of the buffalo farms in the Marmara Region of Turkey.

Indicators	Sub-Groups	N	Relative Frequency
Owner gender	Male	48	92.3
	Female	4	7.7
Education status	Literate	1	1.9
	Primary school	29	55.8
	Secondary school	6	11.5
	High school	12	23.1
	Vocational school	3	5.8
	University	1	1.9
Grazing information	Communal grazing	34	65.4
	Private grazing	18	34.6
Membership of union breeders	Yes	52	100.0
	No	0	0.0

**Table 3 animals-12-03056-t003:** Eigenvalues obtained from multiple factor analysis and percentages of variance explained.

	MFA Eigenvalues		
	Total	% of Variance	Cumulative % of Variance
Dimension 1	2.7	19.1	19.1
Dimension 2	1.7	12.1	31.2
Dimension 3	1.5	10.1	41.3
Dimension 4	1.3	9.2	50.5
Dimension 5	1.2	8.5	59.0

**Table 4 animals-12-03056-t004:** Factor loadings for original indicators after varimax-rotated extracted components.

		Factor Interpretation *					
Indicators	Communality	Milk Productivity	Economic Efficiency	Roughage Management	Dual Purpose Production	Concentrate Intake	Fodder Production
Land used for roughage production, ha	0.85	0.15	0.02	−0.10	−0.01	−0.03	**0.88**
Land used for roughage and grain production per breeding buffalo, ha	0.82	−0.10	0.02	−0.16	0.04	0.07	**0.83**
Highest number of lactating buffalo/Total number of breeding buffalo	0.86	0.19	**0.72**	0.19	−0.31	0.28	−0.12
Number of buffalo calves	0.81	**0.67**	−0.01	−0.17	−0.11	−0.17	−0.06
Concentrate feed offer/Roughage offer in fall–winter/Lactating buffalo	0.79	0.11	−0.03	0.14	−0.07	**0.53**	−0.48
Concentre feed offer per 1 L of milk production, kg	0.72	0.03	**−0.63**	−0.02	0.31	0.35	−0.12
Days in milk, day	0.90	**0.75**	0.16	0.04	−0.16	0.02	−0.16
Average milk production in fall–winter, L	0.89	**0.78**	0.22	−0.12	−0.24	0.13	−0.20
Average milk production in spring–summer, L	0.86	**0.76**	0.16	0.10	−0.19	0.24	−0.23
Milk price, USD	0.79	0.33	−0.14	**0.60**	−0.02	0.04	−0.24
Share of dairy income in total income, %	0.90	0.09	0.08	−0.09	**−0.93**	0.04	−0.03
Gross production value/1 L milk production, USD	0.94	−0.11	0.06	−0.28	**0.84**	0.19	0.06
Variable costs/Breeding buffalo, USD	0.93	0.25	−0.04	−0.39	−0.04	**0.84**	0.12
Variable costs/1 L of milk production, USD	0.96	0.05	**−0.60**	−0.31	0.49	0.41	0.03
Concentrate costs/Breeding buffalo, USD	0.96	0.04	−0.08	0.01	0.00	**0.94**	−0.05
Concentrate costs/1 L of milk production, USD	0.87	−0.14	−0.48	0.10	0.42	**0.65**	−0.08
Roughage costs/Breeding buffalo, USD	0.83	−0.04	−0.06	**−0.83**	−0.06	0.16	0.19
Roughage cost/1 L of milk production, USD	0.93	−0.21	−0.40	**−0.67**	0.29	−0.01	0.17
Gross profit/AWU, USD	0.82	0.24	**0.56**	0.16	0.05	−0.06	0.10
Gross profit/Breeding buffalo, USD	0.94	−0.02	**0.89**	−0.11	0.30	−0.13	0.06
Gross profit/1 L of milk production, USD ^#^	0.94	−0.19	**0.74**	−0.02	0.54	−0.21	0.04
Family labour/Total labour, AWU	0.96	**−0.90**	0.12	−0.19	−0.02	−0.07	−0.19
Paid labour/Total labour, AWU	0.96	**0.90**	−0.12	0.19	0.02	0.07	0.19
Purchased-to-total feed cost, %	0.75	0.28	−0.16	0.15	−0.04	0.47	**−0.50**
Fattening calves’ income/Dairy income, %	0.93	−0.11	0.08	0.06	**0.95**	−0.01	−0.01
% of variance		14	12	12	11	9	8

* Loadings greater than 0.50 were considered significant for the interpretation of the factor definition, and indicators were written in bold to represent corresponding cluster. ^#^ This indicator was double-loaded into economic efficiency and dual-propose farming clusters.

**Table 5 animals-12-03056-t005:** The effect of region, investment level, grazing type, milk productivity, and profitability on the 6 extracted components.

	Province ^1^				Investment Level ^2^				Grazing Type ^3^			Milk Production ^4^				Profitability ^5^			
	IST	KOC	BAL	*P*	L	M	H	*P*	OWN PAS	COM	P	L	M	H	P	L	M	H	P
Milk productivity	3.89 ^a^	−3.93 ^b^	−3.04 ^b^	<0.001	−2.33 ^b^	−1.59 ^b^	4.46 ^a^	<0.001	−3.00	1.59	0.003	−5.23 ^c^	−0.67 ^b^	6.58 ^a^	<0.001	−4.34 ^b^	0.32 ^a^	3.70 ^a^	<0.001
Economic efficiency	0.42	0.54	−1.97	0.26	0.53	−1.08	0.52	0.44	−0.14	0.07	0.86	−2.09 ^b^	−0.24 ^ab^	2.57 ^a^	0.01	−3.39 ^c^	−0.55 ^b^	4.49 ^a^	<0.001
Roughage management	−2.29 ^b^	2.52 ^a^	1.43 ^a^	<0.001	−0.76	1.37	−0.56	0.10	2.49	−1.32	<0.001	1.38	0.05	−1.47	0.07	1.13	0.31	−1.73	0.06
Dual purpose production	−0.94	0.53	1.46	0.27	0.21	0.55	−0.83	0.63	0.61	−0.32	0.46	2.64 ^a^	0.37 ^a^	−3.38 ^b^	<0.001	0.57	−0.001	−0.56	0.81
Concentrate intake	1.43 ^a^	−2.66 ^b^	0.94 ^a^	0.002	−1.62	0.98	0.88	0.09	−2.14	1.13	0.004	−1.75	0.47	0.81	0.19	−0.63	0.17	0.30	0.81
Fodder production	−1.73 ^b^	2.30 ^a^	0.42 ^ab^	<0.001	−0.59	0.49	0.18	0.62	3.56	−1.89	0.001	1.65	−0.11	−1.44	0.07	1.00	−0.06	−0.88	0.38

^a–c^ Values with different superscript letters in a column are significantly different at *p* < 0.05. ^1^ IST refers to Istanbul, Koc refers to Kocaeli, and Bal refers to Balikesir. ^2^ L refers to low scores (0–5), M refers to medium scores (6–7), and H refers to high scores (8–10) of investments in buildings and equipment. ^3^ OWN PAS refers to own pasture, COM refers to common grazing. ^4^ L refers to low, <I quartile = 4.43; M refers to medium between 4.44 and 5.44; and H refers to high > III quartile 5.45 of the total milk production. ^5^ L refers to low, <I quartile = 14, M refers to medium between 15 and 39, and H refers to high > III quartile = 40 of the gross profit.

## Data Availability

Data available on request due to the privacy of the farms.

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
