# Peer review of "Characteristics of Buffalo Farming Systems in Turkey Based on a Multivariate Aggregation of Indicators: A Survey Study"

_animals, 2022, doi:10.3390/ani12213056_

Round 1

Reviewer 1 Report

Comments to authors

The article describing the typology of buffalo farming systems in Turkey was interesting. Instead of descriptive analysis, the authors used the multivariate aggregate model (inferential analysis) to present the characteristics of buffalo farms based on location, intensification, feeding management and economics. Although, the manuscript has a limited scope as for as its application of results is concerned, but the study does have a potential to encourage researchers to replicate the method for performance benchmarking in other regions.   

Title:

The word “classification” is somewhat misleading. The manuscript did not define new farming systems instead presented different attributes of farming systems in Turkey. So, use some other appropriate words e.g., “Characteristics of buffalo faming systems in Turkey based on multivariate aggregation indicators: A survey Study”.          

Summary:

Summary was easy to understand.

Abstract:

The abstract needs some improvement.

Make sure the abstract has 200 words at maximum.  

Line 26-27: Please clarify the sentence.

L 30-31: There is no need to use quotation marks for the listed components.

L 33-37: Rephrase the sentences and use the active voice for clarity of results. It is difficult to understand which variable is related to which.

L 38-39: Usually, observational studies are not that conclusive. It is better just to conclude the association. Example: The results conclude that productivity and profitability were associated with intensive management of buffaloes.

Introduction:

L 43-45: Please start the paragraph with the description of Turkey’s data instead of Italy.

L 46-51: Avoid comparing the Turkish farms with the Italian farms. Just present the Turkish farms.

L 64-68: Delete these sentences. These are not adding anything to build the introduction.

L 77-80: A very long sentence; concise it please.

L 83-98: This is the statistical description. It could be briefly described in the material and methods section. Please remove these lines from here.

L 105-110: Place this paragraph after the sentence ending in line 100.

Materials and Methods:

This section provided adequate description required to repeat the study.

Line 33: face-to-face interviews.

Line 143-147: The quotation marks seemed distracting. The grouping of variables is understandable without these marks.

Line 164: Here the brief description of multiple factor analysis can be provided.

Line 165-166: This sentence needs to be rephased for clarity.

Results:

L 203-223: If possible, the data for these variables can be provided as a supplementary file.

L 253:  Explain all the abbreviations in the caption including dim and hcpc.

L 267: Please describe the positively and negatively loaded variables, preferably in the material and methods section.

The cluster naming was not relevant to the variables they depict. For example, cluster 1 is determined by number of buffalo calves, days in milk, and average milk production; to categorize this cluster as intensification is misleading. If it was done due to the variable “labor types” then better to separate milk production from labor type.  Come up with more relevant and simple naming for all the clusters. “Self-feed” could be replaced with “fodder production”, “intensification” with “milk production”. The authors can also increase or decrease the cluster groupings for better classification of variables.

L 305-307:  Higher investment level were related to higher milk yield. The use of term “intensification” was misleading and created ambiguity.

L 355: Was it “pasture system” or “pastoral” meaning pastoralism?

Discussion:

Overall, the discussion was well organized. Moderate English editing would strengthen this section.

Conclusion:  

The conclusion was relevant to the study. However, after regrouping the variables, as suggested previously, the description of results and the conclusion would slightly be changed. The suggestions can be presented in the next paragraph starting from line 505.

Author Response

Comments of reviewers on first version of manuscript Status Place of changes on new version of manuscript / Explanations

The word “classification” is somewhat misleading. The manuscript did not define new farming systems instead presented different attributes of farming systems in Turkey. So, use some other appropriate words e.g., “Characteristics of buffalo faming systems in Turkey based on multivariate aggregation indicators: A survey Study”.

done The title was changed in accordance with reviewer’s suggestion.
Summary was easy to understand.   Thank you.

The abstract needs some improvement.

Make sure the abstract has 200 words at maximum.  

Line 26-27: Please clarify the sentence.

done Abstract was rewritten. Line 26-27 was updated.
L 30-31: There is no need to use quotation marks for the listed components. done Quotation marks were deleted.
L 33-37: Rephrase the sentences and use the active voice for clarity of results. It is difficult to understand which variable is related to which.   Please see lines 38-41.
L 38-39: Usually, observational studies are not that conclusive. It is better just to conclude the association. Example: The results conclude that productivity and profitability were associated with intensive management of buffaloes. done Example sentence given by the reviewer was used to conclude. Please see lines 43-46.
L 43-45: Please start the paragraph with the description of Turkey’s data instead of Italy. Explained/done Italy is the reference country for intensive buffalo husbandry and the reason of starting introduction with the Italian buffalo husbandry is giving reader an understanding of an intensive buffalo husbandry application and its outcome as well as Turkey’s production potential as a significant buffalo milk producer for the region but its lower production traits.
L 46-51: Avoid comparing the Turkish farms with the Italian farms. Just present the Turkish farms. explained As requested by the reviewer, we started the introduction with the Turkish buffalo stats, however we would like to keep comparing Turkish buffalo husbandry with the intensive Italian buffalo sector for the reasons above mentioned.
L 64-68: Delete these sentences. These are not adding anything to build the introduction. done Sentences were deleted.
L 77-80: A very long sentence; concise it please. done The sentence was shortened. Please see lines 86-88.
L 83-98: This is the statistical description. It could be briefly described in the material and methods section. Please remove these lines from here. done This paragraph was placed under M&M section, Multivariate Analysis subsection. Please see L 201-216.
L 105-110: Place this paragraph after the sentence ending in line 100. done The paragraph was placed as requested. L 111.
This section provided adequate description required to repeat the study.   Thank you.
Line 33: face-to-face interviews. done Please see L 154.
Line 143-147: The quotation marks seemed distracting. The grouping of variables is understandable without these marks. done Please see L 164-168.
Line 164: Here the brief description of multiple factor analysis can be provided. done Please see L 202-216.
Line 165-166: This sentence needs to be rephased for clarity. done Please see L 222-223.
L 203-223: If possible, the data for these variables can be provided as a supplementary file. done A supplementary table was provided. Please see Table A2.
L 253:  Explain all the abbreviations in the caption including dim and hcpc. done We updated the figure 1. Explanations were provided in the updated version. Please see line 316-320 and L 322-327.
L 267: Please describe the positively and negatively loaded variables, preferably in the material and methods section.   Explanations were provided as in M&M section as well as in a supplementary table. Please see L 169-177 and Table A1.
The cluster naming was not relevant to the variables they depict. For example, cluster 1 is determined by number of buffalo calves, days in milk, and average milk production; to categorize this cluster as intensification is misleading. If it was done due to the variable “labor types” then better to separate milk production from labor type.  Come up with more relevant and simple naming for all the clusters. “Self-feed” could be replaced with “fodder production”, “intensification” with “milk production”. The authors can also increase or decrease the cluster groupings for better classification of variables. done/explained We thank reviewer for his suggestion regarding renaming component 1 and component 6. Yes, we agreed that describing this component as “milk productivity” would be more suitable because concentrate offer didn’t load here. Also, we changed self-production as fodder production for the component 6. New descriptions of the components were applied throughout the text.We discussed the descriptions of other components, however we think they finely correspond to the loaded indicators. When choosing the components, we used KMO value higher than 0.6, therefore increasing or decreasing components is not possible.
L 305-307:  Higher investment level were related to higher milk yield. The use of term “intensification” was misleading and created ambiguity. done The term intensification was changed with milk productivity. Please see L 381-383.
L 355: Was it “pasture system” or “pastoral” meaning pastoralism? done We meant the small-scale farms. Please see L 433.
Overall, the discussion was well organized. Moderate English editing would strengthen this section.   Thank you.
The conclusion was relevant to the study. However, after regrouping the variables, as suggested previously, the description of results and the conclusion would slightly be changed. The suggestions can be presented in the next paragraph starting from line 505. done The conclusion was rewritten considering renaming of the component 1. Please see L 594-637.

Reviewer 2 Report

The article is very interesting and shows a very solid analysis using Principal Components on evaluation of dairy buffaloes’ production systems in Turkey.

The article needs some adjustments to be able to publication as follows:

ABSTRACT

 Your abstract need to show the main results that supported your conclusion.

 INTRODUCTION

 Reference number 5 must be changed by one that´s official or presented on a peer reviewed vehicle. I tried to access the file but it was not possible to me. Maybe you can use reference number 7 for the same purpose.

 Lines 103 to 104 – this kind of information, referring to your own study, should not be used on the introduction, you can use this on your discussion.

 MATERIAL AND METHODS

 Line 134 – the corresponded production period must be checked, if data collection was performed in June and December 2018, how can you use the production period 2018 to 2019? If your information is correct, please explain.

 Data collection: your database collection is poorly described, tables 1a doesn´t show the methodology used to estimate most of the indicators pointed, especially Animal Production and Economic performance information. Some indicators must be defined on table footer.

It was not clear how you find the Economic information; you must show if the producers gave you all the parameters to make the estimates or if you use standard indicators to estimate these parameters.

You must show the complete questionnaire applied to the farmers as a supplementary material, maybe you can do the same to give a better description of the equations and other methodologies used to compose your database.

 Seasons of the year – you must describe the clima to justify only two seasons of the year (winter and summer) and to characterize them. Winter and summer duration is the same (6 months each)?

 Animal Nutrition information:

 Consumptions (concentrate and forage) are on dry matter bais? If not you must make it clear.

You must describe animal category, specially on weight or metabolic weight basis.

You must describe consumption, because maybe “offer” is the term that best explain this parameter (I think you didn´t measured feed consumption).

Annual pasture use – please define this parameter. What is the difference of Annual use and Pasture use in winter and summer? If I sum pasture in winter + pasture in summer = 362, compared to total use (2,181.0), we find a proportion of only 16% of total value, does it mean that pasture use is more relevant on spring and fall?

 I pointed other observations on table 1a on the document attached.

 RESULTS

 Line 205 – “Some herds used communal grazing, while others used their owned land for grazing purposes.” If this is an important information to describe production system you must show the proportion (%) of properties that use this management strategy. It is important to show the areas disponible for communal grazing (did you considered this area to evaluate properties efficiency or only the owned land? This is an important information and must be clear in the article).

 Line 230 – how did you estimate total DM intake, did you included pasture intake (even estimated) to determine proportion of silage and concentrate on diet? Tis methodology must be showed on Material and Methods.

 Line 232 – did you correct milk production for a standard lactation duration to compare different herds? This is not clear on material and methods.

 Line 241-242 - Please show the proportion of farmers with 0 values. I think it is important to show the proportion of farms on different stratification (maybe as supplementary material).

 Figure 1 – The figure must better elaborated, quartes and dimension are not clear. I think a legend is necessary (please read the observations on the text attached).

 Lines 264-291 – this is a very long paragraph, please divide it into 2 or 3 paragraphs.

 Table 3 – need some adjustments (please read the observations on the text attached).

 DISCUSSION

 Lines 416-417 - Can you estimate the area/animal disponible for this porupose? Is it considered on the evaluation?

 Lines 427-428 – I can´t make comments about economic efficiency because you didn´t detailed economic analysis on material and methods

 Lines 448-470 – in this paragraph you discuss contradictory findings regarding the effect of intensification on profitability, I think you must put on your discussion the low productive potential from the dairy buffaloes used on the production system, with this limitation, concentrate use generally is less profitable than roughage.

 Figures 2a and 2b – please, read the comments on the text attached.

 CONCLUSIONS

 Line 499-501 - Please exclude from your conclusion the topic about “sustainability”. You did not evaluated sustainability on your study, so this information can only be used on your discussion as inference from articles found on literature.

Author Response

Reviewer 2    
Comments of reviewers on first version of manuscript Status Place of changes on new version of manuscript / Explanations
Your abstract need to show the main results that supported your conclusion.   Abstract was rewritten. Please see L 26-46.
Reference number 5 must be changed by one that´s official or presented on a peer reviewed vehicle. I tried to access the file but it was not possible to me. Maybe you can use reference number 7 for the same purpose. Explained/done We thank reviewer for drawing our intention to the references. In fact, we mistakenly used alphabetic order when organizing the reference list instead of using numerical order. In the current reference number 5, we refer to the national buffalo project. Please see the corrected reference list. However, we would like to indicate that previous reference number 5 is an official and trusted source, it was reported by the Ministry of industry and technology, Agency of Directorate of Development.
 Lines 103 to 104 – this kind of information, referring to your own study, should not be used on the introduction, you can use this on your discussion. done This information was excluded from the text.
Line 134 – the corresponded production period must be checked, if data collection was performed in June and December 2018, how can you use the production period 2018 to 2019? If your information is correct, please explain. done The production period was corrected as 2017-2018 in the text. L 155.
 Data collection: your database collection is poorly described, tables 1a doesn´t show the methodology used to estimate most of the indicators pointed, especially Animal Production and Economic performance information. Some indicators must be defined on table footer. done Please see the supplementary table 1 and line 169-177.
It was not clear how you find the Economic information; you must show if the producers gave you all the parameters to make the estimates or if you use standard indicators to estimate these parameters. done Please see the supplementary table 1 and line 160-170.
You must show the complete questionnaire applied to the farmers as a supplementary material, maybe you can do the same to give a better description of the equations and other methodologies used to compose your database. done Please see the supplementary file.
Animal Nutrition information:    
 Consumptions (concentrate and forage) are on dry matter bais? If not you must make it clear. done No, they are in as fed basis. Please see line 174.
You must describe animal category, specially on weight or metabolic weight basis. done Please see line 175-176.
You must describe consumption, because maybe “offer” is the term that best explain this parameter (I think you didn´t measured feed consumption). done Thank you for warning us. Yes, we didn’t estimate consumption. We also think offer is a better term. Upon your suggestion, consumption was changed to offer throughout the text.
Annual pasture use – please define this parameter. What is the difference of Annual use and Pasture use in winter and summer? If I sum pasture in winter + pasture in summer = 362, compared to total use (2,181.0), we find a proportion of only 16% of total value, does it mean that pasture use is more relevant on spring and fall? done Indicators were corrected as Total annual pasture use, average monthly pasture use in fall-winter, average monthly pasture use in spring-summer. Fall-winter consisted of January, February, March, September, October, November, and December. Spring-Summer consisted of April, May, June, July, August, and this was indicated as footnote in Table 1a. Please see Table 1a, L 191.
I pointed other observations on table 1a on the document attached. done Thank you, your suggestions were applied. We would like to indicate that 0 values for the number of buffalo bull was caused by some herds didn’t obtain a bull. They share/change a bull during breeding season among other herds. We indicated this information in results of the general characteristics of the farms. Your question regarding the days in milk in summer and winter: Unfortunately, we don’t have this information. Please see Table 1a, L 191.
 Line 205 – “Some herds used communal grazing, while others used their owned land for grazing purposes.” If this is an important information to describe production system you must show the proportion (%) of properties that use this management strategy. It is important to show the areas disponible for communal grazing (did you considered this area to evaluate properties efficiency or only the owned land? This is an important information and must be clear in the article). done/explained We have provided a supplementary table showing the percentages of the farms using different grazing types. Please see the Table S2. We also would like to make it clear, except for the owned land, other areas including, meadow, forest are open for public use for grazing. Therefore, any herd (i.e., cattle, sheep, goat) are allowed to have access to those areas. The number of grazing animals in publicly open grazing lands was not possible to obtain. For that reason, we didn’t evaluate properties efficiency.
 Line 230 – how did you estimate total DM intake, did you included pasture intake (even estimated) to determine proportion of silage and concentrate on diet? Tis methodology must be showed on Material and Methods. done Please see lines 174-175.
 Line 232 – did you correct milk production for a standard lactation duration to compare different herds? This is not clear on material and methods. done Information was provided. Please see lines 171-173.
 Line 241-242 - Please show the proportion of farmers with 0 values. I think it is important to show the proportion of farms on different stratification (maybe as supplementary material). done A supplementary table was provided. Please see Table A3.
 Figure 1 – The figure must better elaborated, quartes and dimension are not clear. I think a legend is necessary (please read the observations on the text attached). done Figure 1 was updated. Please see figure 1a and figure 1b. L315 and L321.
Lines 264-291 – this is a very long paragraph, please divide it into 2 or 3 paragraphs. done This paragraph was divided into 6 paragraphs in which a cluster is defined. L 335-365.
 Table 3 – need some adjustments (please read the observations on the text attached). done Your suggestions were addressed. Please see table 3. L 368.
Lines 416-417 - Can you estimate the area/animal disponible for this porupose? Is it considered on the evaluation? explained We only have that information for farms in Kocaeli where they used their own land for grazing. Farmers didn’t provide trusted data for the available size of meadow and forest as well as number of animals grazing on them. Therefore, we didn’t include area/animal indicator in the analysis. 
 Lines 427-428 – I can´t make comments about economic efficiency because you didn´t detailed economic analysis on material and methods done Details about calculation of economic indicators were provided in Table S1.
 Lines 448-470 – in this paragraph you discuss contradictory findings regarding the effect of intensification on profitability, I think you must put on your discussion the low productive potential from the dairy buffaloes used on the production system, with this limitation, concentrate use generally is less profitable than roughage. Explained/done The paragraph was reorganised due to the change in first cluster definition. We thank reviewer for his comment. An explanation to the relationship between low milk potential of the buffaloes, concentrate and roughage intake, as well as their economic impact was provided. Please see L 557-560.
 Figures 2a and 2b – please, read the comments on the text attached. done Figures were updated considering the new descriptions as well as your suggestions were applied. L 584 and L589.
Line 499-501 - Please exclude from your conclusion the topic about “sustainability”. You did not evaluated sustainability on your study, so this information can only be used on your discussion as inference from articles found on literature. done This sentence was re-written. Please see lines 607-610.

Reviewer 3 Report

Manuscript ID: animals-1985434

Article

Classification of buffalo farming systems in Turkey based on a multivariate aggregation of indicators gathered from farmers’ interview

General remarks

The authors propose a manuscript titled “Classification of buffalo farming systems in Turkey based on a multivariate aggregation of indicators gathered from farmers’ interview”. The article is interesting and well-argued in some aspects of Turkish dairy farming, where buffaloes are vital for the local rural economies. Following the idea of the authors, understanding farm heterogeneity is critical to identify targeted interventions that have the potential to improve technical efficiency and sustainable intensification of buffalo farming. While interesting, I believe the manuscript needs careful revision in several parts. My perplexities mainly concern the introduction and, at least in part, the description of the materials and methods. Therefore, I have suggested a major revision. Below are indicated some comments useful for further improving the manuscript. After these few suggestions, the work may be published. Hoping to have contributed to improving the manuscript's quality. Good work.

Specific comments

Title: please decline the word "interview" in the plural. Thanks. 

Throughout the manuscript: Authors are invited to carefully check the manuscript' formatting scheme. For example, according to the journal template, the text must be justified, and, within each paragraph, no breaks are expected (such as between lines 63-64, 76-77, etc.). Furthermore, the font used in some tables (for example Table 3) does not seem to coincide with the recommended one. Thanks.

L 16-17: In my opinion, in defining the manuscript aims the authors should indicate the geographic area to which the study refers, and, within these, toward which farms the research focuses (i.e., buffalo herds). Thanks.

L 40 (Keywords): in my opinion, I believe it is useful to replace “buffalo” with “buffalo farms”. Thanks.

L 43-53: In the Mediterranean basin and, more generally in Europe, the Italian buffalo sector represents the reference point to describe buffalo farming intensification. Given the purpose of the research, I believe it is correct that the authors refer to it. However, in this regard, I believe that some changes are needed, mainly with regard to the references. First, to support the claim in lines 43-45 (i.e., "In Europe and Eurasia, Italy is the leading country with its buffalo milk production where the high production is obtained with specialized breed under intensive conditions”) the authors can refer to doi.org/10.3390/ani10030515, doi.org/10.3390/agriculture12081219, and doi.org/10.3168/jds.2017-14157 for updating the text. Reference [2] (i.e., Abdi and Williams, 2010) deals with the principal component analysis and does not define, from what I have read, the milk production of Italian or Turkish buffaloes. Therefore, I suggest updating. Referring to the Italian dairy buffalo sector, the authors can add that, unlike the Turkish one, the production of cheese in Italy is mainly devoted to pasta filata cheese production. Thus, the sentence could be rearranged as: "In contrast to Italian farms, where the milk is entirely devoted to pasta filata cheeses production [doi.org/10.1111/1471-0307.12640 and doi.org/10.3390/molecules25061332 as references], Turkish buffalo farming is conducted in pasture-based systems by small/medium scale family farms with low-yield Anatolian breeds whose milk is processed into yogurt or kaymak (milk cream) [3]”. Alvarez et al. (2008) that the authors cite (reference [4] in the text; line 52) analyze the relationships between intensification and efficiency, but do not consider buffaloes farming; moreover, from what I have read, the same paper does not provide data referring to 2018 (having been published in 2008). Therefore, authors are advised to correctly update the reference [4]. The sentence on lines 52-53 ("However, when considering the milk production, the reported buffalo milk production was only 0.4% of that of large ruminants") should be supported by a reference, in my opinion. Thanks.

L 65-66: what do the authors mean by "output orientation"? the sentence is, in my opinion, unclear. Authors are encouraged to paraphrase. Thanks.

L 77-80: authors are advised to avoid redundancy (i.e., “change” and “changes”). Please, re-phrase. Thanks.

L 100: authors are certain that [26] is the right reference to support the statement made? Please check, thanks.

L 101-104: In my opinion, the introduction should provide the information necessary to outline the problem and how the designed research contributes to solving it (i.e., the aims). Therefore, I believe that this part of the text (in particular "Typology studies has recently started to be included in animal husbandry context [25] in Turkey, to our knowledge current study will be the first study that is clustering the buffalo farms based on their technical and economic parameters ") can be more usefully re-proposed in discussions. Thanks.

L 105: in my opinion, it would be appropriate to specify which factors the previous studies referred to by the authors refer to. Thanks.

L 127: to provide the dimension of the farms' panel potential available, authors could provide, if possible, the statistics on the total farms located in the study area. Thanks.

L 133: please, replace “survey” with “interview”. Thanks.

L 134: the authors will forgive me, but some accounts don't add up! If the interviews with the farmers were carried out in the period June-December 2018, how can the collected data correspond to the 2019 production period? Perhaps the authors wanted to refer to 2017-2018. Please verify. Thanks.

L 137: What do the authors mean by "farmer profile topics? Explain, please. Thanks.

L 137 -142: it’s unclear enough to me how the authors calculate the investment in building and equipment scores, and what criteria they used to assign a score from 0 to 10 (what 0, 1, 2, ... corresponds to). Thanks.

L 149: check the formatting of the sub-paragraphs (see the first comment). Thanks.

L 203-205: the climatic characteristics to which period do they refer? Are they averages of historical data (if yes, indicate to which time frame they refer and which is the source of the data) or of values recorded by the authors (if yes, indicate the methods used in the appropriate paragraph)? Thanks.

L 203-223: there is a complete lack of references to tables and graphs as if they were discussions rather than results.

L 210: the red dog is a local variety of wheat? The authors are requested to clarify, thanks.

L 359-361: since the one described is a phenomenon also common to productive realities different from those to which the authors refer, in my opinion, the sentence could be supported by a reference. In this regard, I suggest doi.org/10.3168/jds.2018-14710. Thanks.

L 367-370: I probably misunderstood. If so, I apologize in advance to the authors. Normally, agriculture (in its broadest sense) is considered a declining sector compared to other productive activities. Therefore, the workforce tends to move from the agricultural sector to the industrial one, which guarantees a higher wage for the laborer. It is not clear to me, therefore, how a more developed and industrialized province can provide cheap labor for agricultural activities.

L 401-404: the strong competition with non-zootechnical agricultural activities (horticulture and floriculture, among others) and urbanization are among the main drivers of the intensification of the buffalo sector in other areas. Authors could create useful parallelisms, referring, for instance, to doi.org/10.1017/S0021859618000072.

L 429: it doesn't seem to me that dual-purpose farms have been defined previously. Please check. Thanks.

L 471-472: in my opinion, tables and figures should not be recalled in discussions. Thanks.

L 479: the study by Atzori et al. does not correspond to reference [10] but to 6. I ask the authors to carefully check the sequence of references in the text and update the bibliography list accordingly. Thanks.

Author Response

Reviewer 3    
Comments of reviewers on first version of manuscript Status Place of changes on new version of manuscript / Explanations
Title: please decline the word "interview" in the plural. Thanks.  done The title was changed as requested by the reviewer 1st.
Throughout the manuscript: Authors are invited to carefully check the manuscript' formatting scheme. For example, according to the journal template, the text must be justified, and, within each paragraph, no breaks are expected (such as between lines 63-64, 76-77, etc.). Furthermore, the font used in some tables (for example Table 3) does not seem to coincide with the recommended one. Thanks. done Thank you for warning. We addressed your concern.
L 16-17: In my opinion, in defining the manuscript aims the authors should indicate the geographic area to which the study refers, and, within these, toward which farms the research focuses (i.e., buffalo herds). Thanks. done Please see line 16-17.
L 40 (Keywords): in my opinion, I believe it is useful to replace “buffalo” with “buffalo farms”. Thanks. done Please see line 47.
L 43-53: In the Mediterranean basin and, more generally in Europe, the Italian buffalo sector represents the reference point to describe buffalo farming intensification. Given the purpose of the research, I believe it is correct that the authors refer to it. However, in this regard, I believe that some changes are needed, mainly with regard to the references. First, to support the claim in lines 43-45 (i.e., "In Europe and Eurasia, Italy is the leading country with its buffalo milk production where the high production is obtained with specialized breed under intensive conditions”) the authors can refer to doi.org/10.3390/ani10030515, doi.org/10.3390/agriculture12081219, and doi.org/10.3168/jds.2017-14157 for updating the text. Reference [2] (i.e., Abdi and Williams, 2010) deals with the principal component analysis and does not define, from what I have read, the milk production of Italian or Turkish buffaloes. Therefore, I suggest updating. Referring to the Italian dairy buffalo sector, the authors can add that, unlike the Turkish one, the production of cheese in Italy is mainly devoted to pasta filata cheese production. Thus, the sentence could be rearranged as: "In contrast to Italian farms, where the milk is entirely devoted to pasta filata cheeses production [doi.org/10.1111/1471-0307.12640 and doi.org/10.3390/molecules25061332 as references], Turkish buffalo farming is conducted in pasture-based systems by small/medium scale family farms with low-yield Anatolian breeds whose milk is processed into yogurt or kaymak (milk cream) [3]”. Alvarez et al. (2008) that the authors cite (reference [4] in the text; line 52) analyze the relationships between intensification and efficiency, but do not consider buffaloes farming; moreover, from what I have read, the same paper does not provide data referring to 2018 (having been published in 2008). Therefore, authors are advised to correctly update the reference [4]. The sentence on lines 52-53 ("However, when considering the milk production, the reported buffalo milk production was only 0.4% of that of large ruminants") should be supported by a reference, in my opinion. Thanks. done We thank reviewer for drawing our intention to the references. In fact, we mistakenly used alphabetic order when organizing the reference list instead of using numerical order. We corrected the list. We appreciate the recommended articles. We referred them to explain intensive buffalo management in Italy and also product differences. We referred to the national statistical institute for the share of buffalo milk. Please see 61-62.
L 65-66: what do the authors mean by "output orientation"? the sentence is, in my opinion, unclear. Authors are encouraged to paraphrase. Thanks. done This paragraph was excluded from the text as suggested by reviewer 1st.
L 77-80: authors are advised to avoid redundancy (i.e., “change” and “changes”). Please, re-phrase. Thanks. done The sentences were rephrased. Please see L 86-88.
L 100: authors are certain that [26] is the right reference to support the statement made? Please check, thanks. done Thank you for your warning, it was corrected.
L 101-104: In my opinion, the introduction should provide the information necessary to outline the problem and how the designed research contributes to solving it (i.e., the aims). Therefore, I believe that this part of the text (in particular "Typology studies has recently started to be included in animal husbandry context [25] in Turkey, to our knowledge current study will be the first study that is clustering the buffalo farms based on their technical and economic parameters ") can be more usefully re-proposed in discussions. Thanks. done We excluded mentioned part upon your request.
L 105: in my opinion, it would be appropriate to specify which factors the previous studies referred to by the authors refer to. Thanks. done Please see line 110-114.
L 127: to provide the dimension of the farms' panel potential available, authors could provide, if possible, the statistics on the total farms located in the study area. Thanks. done Total number of buffalo farms in each province were provided. This information was obtained through Buffalo Breeders Association. Please see L146-148.
L 133: please, replace “survey” with “interview”. Thanks. done Please see L 154.
L 134: the authors will forgive me, but some accounts don't add up! If the interviews with the farmers were carried out in the period June-December 2018, how can the collected data correspond to the 2019 production period? Perhaps the authors wanted to refer to 2017-2018. Please verify. Thanks. done Please see L 155.
L 137: What do the authors mean by "farmer profile topics? Explain, please. Thanks. done Please see L 134.
L 137 -142: it’s unclear enough to me how the authors calculate the investment in building and equipment scores, and what criteria they used to assign a score from 0 to 10 (what 0, 1, 2, ... corresponds to). Thanks. explained For each equipment owned by the farmer, 1 point was given, and the total score was obtained by summing. This was explained in line 158-164.
L 149: check the formatting of the sub-paragraphs (see the first comment). Thanks. done  
L 203-205: the climatic characteristics to which period do they refer? Are they averages of historical data (if yes, indicate to which time frame they refer and which is the source of the data) or of values recorded by the authors (if yes, indicate the methods used in the appropriate paragraph)? Thanks. done Yes, they are the averages of the historical data. Source was provided. Please see L 262.
L 203-223: there is a complete lack of references to tables and graphs as if they were discussions rather than results. done A table was provided as supplementary. Please see Table A2.
L 210: the red dog is a local variety of wheat? The authors are requested to clarify, thanks. explained No, red dog is not a local variety. The most common milling by-products are from wheat with lesser amounts derived from corn. These are classified on their fibre content and include bran, middlings, shorts, red dog and wheat-germ meal. Source: https://animalbiosciences.uoguelph.ca/~gking/Ag_2350/nutrition.htm
L 359-361: since the one described is a phenomenon also common to productive realities different from those to which the authors refer, in my opinion, the sentence could be supported by a reference. In this regard, I suggest doi.org/10.3168/jds.2018-14710. Thanks. done We thank reviewer for the suggestion. Please see L 439.
L 367-370: I probably misunderstood. If so, I apologize in advance to the authors. Normally, agriculture (in its broadest sense) is considered a declining sector compared to other productive activities. Therefore, the workforce tends to move from the agricultural sector to the industrial one, which guarantees a higher wage for the laborer. It is not clear to me, therefore, how a more developed and industrialized province can provide cheap labor for agricultural activities. explained Yes, we agree that labour wages should be higher in parallel to the industrial development. However, we should also indicate that Istanbul is a very cosmopolite city. There are various unregistered immigrant workers who work for little amount of money. Therefore, we have discussed Istanbul as having more opportunity to find low-cost labour. However, we excluded this part from the discussion due to slightly changed description of cluster 1.
L 401-404: the strong competition with non-zootechnical agricultural activities (horticulture and floriculture, among others) and urbanization are among the main drivers of the intensification of the buffalo sector in other areas. Authors could create useful parallelisms, referring, for instance, to  doi.org/10.1017/S0021859618000072. done We thank reviewer for the suggestion. Please see L 490.
L 429: it doesn't seem to me that dual-purpose farms have been defined previously. Please check. Thanks. explained It was defined in results section. Please see 351-356.
L 471-472: in my opinion, tables and figures should not be recalled in discussions. Thanks. explained We thank reviewer for the comment. Our aim by providing the spider chart is to strengthen discussion rather than using it as a result of study. We think this is optional. We would like to keep it as it is.
L 479: the study by Atzori et al. does not correspond to reference [10] but to 6. I ask the authors to carefully check the sequence of references in the text and update the bibliography list accordingly. Thanks. done Many thanks for warning us. References was checked and corrected.

Round 2

Reviewer 1 Report

It appears that the authors have satisfactorily addressed the technical comments. Some of the changes made in the manuscript could not be confirmed as the line numbers, referenced in the response letter, did not match to the revised version. For example, the conclusion section started from line 514 in the revised manuscript while it was referenced as line 584 in the response letter. The revised manuscript is technically sound; however, it needs adequate corrections in grammatical and punctuation errors. I strongly recommend the manuscript for English editing. It will certainly improve the language quality.

 Some minor comments are listed below for further improvement.

Line 28: Please end the sentence at fodder production. The clause “explained the 82% of the variance” either should be removed or be used in a separate sentence.

Line 36-37: These lines lack clarity in meaning. Please rephrase it.

Line 56-59: Please shorten this sentence or divide it for the readers’ ease. Follow this comment for every sentence longer than 3 lines. The readers may find it difficult to hold the idea in a longer sentence.

Author Response

  Comments of reviewer on  second version of manuscript Status Place of changes on new version of manuscript / Explanations

It appears that the authors have satisfactorily addressed the technical comments. Some of the changes made in the manuscript could not be confirmed as the line numbers, referenced in the response letter, did not match to the revised version. For example, the conclusion section started from line 514 in the revised manuscript while it was referenced as line 584 in the response letter. The revised manuscript is technically sound; however, it needs adequate corrections in grammatical and punctuation errors. I strongly recommend the manuscript for English editing. It will certainly improve the language quality.

explained

We thank reviewer for his/her previous contribution to our manuscript. We think, the reason for the unmatched line numbers (between response letter and the manuscript) might had been caused if the reviewer accepted all track changes while reading the revised manuscript. In the response letter, the line numbers corresponded to numbers without accepting track changes mode.

Our manuscript was edited by Prof Dr Alper Yilmaz who is the Head of Animal Breeding and Husbandry Department in Istanbul University Cerrahpasa Veterinary Faculty and also the Editor of the JIVS (Journal of Istanbul Veterinary Sciences). Besides his academic studies, he has a good command of both written and spoken English (86/100 in KPDS – Turkish Government English Language Exam).

  Line 28: Please end the sentence at fodder production. The clause “explained the 82% of the variance” either should be removed or be used in a separate sentence. done It was removed.

Line 36-37: These lines lack clarity in meaning. Please rephrase it.

done The sentence was rephrased. Line 36.
  Line 56-59: Please shorten this sentence or divide it for the readers’ ease. Follow this comment for every sentence longer than 3 lines. The readers may find it difficult to hold the idea in a longer sentence.   Please see line 58-62. We applied reviewer’s suggestion throughout the text.

Reviewer 3 Report

Dear authors,
I have evaluated the revised version of your manuscript. Based on the changes made, I believe that the manuscript, in its current form, deserves to be published on animals. I give you my best wishes. Congratulations

Author Response

Dear reviewer we thank for all the contributions you have done so far.